# Generalization In Multi-Objective Machine Learning

## Abstract

Modern machine learning tasks often require considering not just one but multiple objectives. For example, besides the *prediction quality*, this could be the *efficiency*, *robustness* or *fairness* of the learned models, or any of their combinations. Multi-objective learning offers a natural framework for handling such problems without having to commit to early trade-offs. Surprisingly, statistical learning theory so far offers almost no insight into the generalization properties of multi-objective learning. In this work, we make first steps to fill this gap: we establish foundational generalization bounds for the multi-objective setting as well as generalization and excess bounds for learning with scalarizations. We also provide the first theoretical analysis of the relation between the Pareto-optimal sets of the true objectives and the Pareto-optimal sets of their empirical approximations from training data. In particular, we show a surprising asymmetry: all Pareto-optimal solutions can be approximated by empirically Pareto-optimal ones, but not vice versa.

## 1 Introduction

Traditionally, statistical machine learning has concentrated on solving one single-objective optimization problem: to minimize the average loss over a given training set. Additional quantities of interest, such as *model complexity*, had to be either addressed implicitly by the choice of model class, or integrated into the main objective via weighted regularization terms. Recently, however, additional quantities of interest have made it into the focus of the machine learning community, such as the *fairness*, *robustness*, *efficiency* or *interpretability* of the learned models. Optimizing these can be in conflict with the goal of low training loss and task-specific trade-offs need to be made. Unfortunately, hard-coding such trade-offs can have undesirable consequences, and model-selecting them is a cumbersome process when multiple objectives are involved.

To avoid the need for *a priori* trade-offs, *multi-objective learning* has recently received increasing attention. Using *multi-objective optimization*, it either finds promising trade-off parameters at the same time as training the actual model, or it computes multiple solutions that reflect different trade-offs, ideally along the complete *Pareto-front*[1] While multi-objective optimization and learning are algorithmically rich fields, their theory is much less well explored. In particular, learning-theoretic results, such as generalization bounds, are almost completely missing.

In this work, we aim at putting multi-objective learning on solid theoretic foundations. Specifically, we present three results of fundamental nature for understanding the properties of learning with multiple objectives. 1) We show that generalization bounds of individual learning objectives carry over also to the situation when learning with multiple objectives simultaneously. 2) We provide generalization and excess bounds that hold uniformly across a broad range of *scalarization* techniques. 3) We analyse in what sense the set of models that are *empirically Pareto-optimal* (i.e. optimal with respect to a training set) approximates the set of models that are actually *Pareto-optimal* (i.e. optimal with respect to the data distribution). Our results provide theoretical justifications for the use of scalarization-based as well as Pareto-based multi-objective optimization in a learning context, though with some caveats that have no analog in single-objective learning.

---

[1]We define the technical terms *Pareto-front*, *Pareto-optimal* and *scalarization* in Section 2.

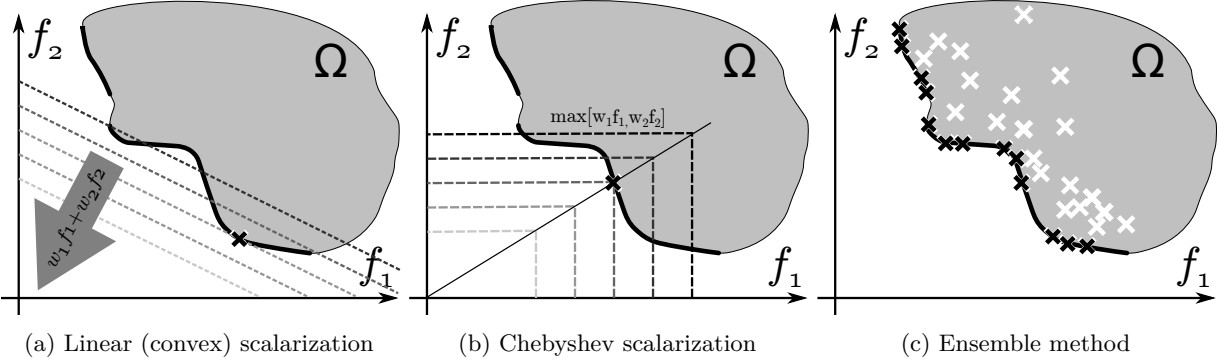

(a) Linear (convex) scalarization    (b) Chebyshev scalarization    (c) Ensemble method

Figure 1: For general multi-objective optimization problems the Pareto-front (bold curves) can be disconnected and non-convex. (a) *Linear scalarization* can find Pareto-optimal solutions on the convex hull of the front. (b) *Chebyshev scalarization* can find solutions everywhere on the front. (c) *Ensemble methods* compute many solutions, aiming for the complete Pareto-front to be represented.

## 2 Notation and background

In this section, we introduce our notation and provide background information on multi-objective optimization and learning, as well as statistical learning theory. Our description follows standard textbooks, such as Miettinen (2012) and Nocedal & Wright (1999) for optimization, and Mohri et al. (2018) and Shalev-Shwartz & Ben-David (2014) for machine learning. More details and derivations can be found there.

### 2.1 Single- and multi-objective optimization

At the heart of most modern machine learning algorithms lies an optimization step. In standard (single-objective) optimization, one is given an input set, $\Omega$, and an objective function, $f : \Omega \to \mathbb{R}$. Because the objective values are just real numbers, they are totally ordered: any two point $\omega, \omega' \in \Omega$ are *comparable* in the sense that at least one of the relations $f(\omega) \leq f(\omega')$ or $f(\omega') \leq f(\omega')$ holds. Consequently, it is a natural question to ask which $\omega^* \in \Omega$ achieve the smallest objective value, if any. A plethora of *single-objective optimization* methods have been developed to answer this question, let it be *gradient-based* (Lemaréchal, 2012; Nocedal & Wright, 1999) or *derivative-free* (Audet & Hare, 2017; Bremermann, 1962).

In *multi-objective optimization*, one is given multiple objective functions, $f_1, f_2, \ldots, f_N : \Omega \to \mathbb{R}$, or equivalently, one vector-valued function, $F : \Omega \to \mathbb{R}^N$ with $F(\omega) = (f_1(\omega), \ldots, f_N(\omega))$. We can again define an associated order relation:

**Definition 1.** For $\omega, \omega' \in \Omega$ we say that $\omega$ *weakly dominates* $\omega'$ if $f_j(\omega) \leq f_j(\omega')$ for all $j \in [N]$. We say $\omega$ *strongly dominates* $\omega'$ if additionally $f_j(\omega) < f_j(\omega')$ for at least one $j \in [N]$.

Because of the multi-dimensional nature, these orderings are only partial. There are pairs $\omega, \omega' \in \Omega$ that are *uncomparable*, i.e. neither $F(\omega) \preccurlyeq F(\omega')$, nor $F(\omega') \preccurlyeq F(\omega)$ holds. Consequently, in multi-objective optimization it typically makes no sense to look for absolute *best* solutions. Instead, one searches for *Pareto-optimal* solutions.

**Definition 2.** A point $\omega^* \in \Omega$ is called *Pareto-optimal* if there is no other point $\omega \in \Omega$ that *strongly dominates* it. The set of all Pareto-optimal points is called *Pareto-optimal set*. The set of corresponding objective value vectors is called *Pareto-front*.

A large number of algorithms have been developed also for multi-objective optimization. When trying to find solutions across the complete Pareto-front, meta-heuristics such as *evolutionary algorithms* (Zitzler & Thiele, 1999) are often employed. If a single Pareto-optimal solution suffices, *scalarizations* in combination with single-objective optimization can be used (Geoffrion, 1968). A *scalarization* function, $\mathcal{U} : \mathbb{R}_+^N \to \mathbb{R}_+$, combines the individual objective values into a single one. Prominent examples are weighted $p$-norms:

$\mathcal{U}_w^{(p)}(x_1, \ldots, x_N) = \left( \sum_{i \in [N]} |w_i x_i|^p \right)^{1/p}$ for $p \in (1, \infty)$, and $\mathcal{U}_w^{(\infty)}(x_1, \ldots, x_N) = \max_{i \in [N]} |w_i x_i|$, where $w \in W \subset \mathbb{R}_+^N$ is a vector of weights that encode a trade-off between the different objectives.

Arguably the most popular choice of scalarization is the $L^1$-norm with weights in the probability simplex $\Delta_N = \{w \in \mathbb{R}_+^N : \sum_i w_i = 1\}$. This means, one forms *convex combinations* of the individual objectives (Gass & Saaty, 1955). For any non-zero choice of weights, minimizers of this resulting scalarized objective will be Pareto-optimal (Geoffrion, 1968). However, the set of solutions obtainable by varying the weights might not recover the complete Pareto-front, unless the optimization problem is convex (Censor, 1977). In contrast, the choice $p = \infty$ (called *weighted Chebyshev norm*) allows recovering the complete Pareto front when varying the weights in $\Delta_N$ (Miettinen, 2012, Chapter 3.4). Figure 1 illustrates these concepts.

## 2.2 Single- and multi-objective learning

Our analysis in this work applies to supervised as well as unsupervised learning. Therefore, we adopt a notation that allows expressing both of these cases in a single concise way. Let $p(z)$ be a fixed but unknown data distribution over a data space $\mathcal{Z}$. We denote by $\mathcal{H}$ a *hypothesis set* and $\ell : \mathcal{Z} \times \mathcal{H} \to \mathbb{R}_+$ a *loss function*. For supervised learning with $\mathcal{H} \subset \{h : \mathcal{X} \to \mathcal{Y}\}$, one uses $\mathcal{Z} = \mathcal{X} \times \mathcal{Y}$, and $\ell(z, h) = L(y, h(x))$, where $L : \mathcal{Y} \times \mathcal{Y} \to \mathbb{R}_+$ measures, e.g., the classification or regression accuracy. For unsupervised learning, one uses $\mathcal{Z} = \mathcal{X}$, and $\ell$ measures, e.g., the reconstruction error of a clustering or dimensionality reduction step.

**Single-objective learning.** Standard (single-objective) learning has the goal of identifying a hypothesis with small *risk* (*expected loss*), $\mathcal{L}(h) = \mathbb{E}_{z \in \mathcal{Z}}[\ell(z, h)]$. To approximate this uncomputable quantity, the learner uses a *training set*, $S = \{z_1, \ldots, z_n\}$ to computes the *empirical risk*, $\widehat{\mathcal{L}}(h) = \frac{1}{n} \sum_{i=1}^n \ell(z_i, h)$.

*Statistical learning theory* studies how well the empirical risk approximates the true risk and under which conditions minimizing the (computable) empirical risk is a good strategy for finding solution with low true risk. Many corresponding results are known. In particular, under well-understood conditions on $\mathcal{H}$ and $S$, one can prove that, with high probability over the sampling of $S$, the true risk is well approximated by the empirical risk, uniformly across all hypotheses. Mathematically, such a guarantee has the form of a *generalization bound*:

$$\forall \delta \in (0, 1) \quad \Pr\left\{ \forall h \in \mathcal{H} : |\mathcal{L}(h) - \widehat{\mathcal{L}}(h)| \leq \mathcal{C}(n, \mathcal{H}, \delta) \right\} \geq 1 - \delta. \tag{1}$$

The problem-dependent *generalization term* $\mathcal{C}(n, \mathcal{H}, \delta)$ typically consists of a *complexity* component that reflects the expressive power of the hypothesis class, and a *confidence* component that reflects the uncertainty due to finite sampling effects. Ideally, both components will converge to 0 when the number of samples grows to infinity.

From bounds of the form (1) one can derive guarantees that, with high probability, solutions obtained by minimizing the empirical risk have close to optimal true risk. Formally, for $\hat{h}^* \in \arg\min_{h \in \mathcal{H}} \widehat{\mathcal{L}}(h)$, an *excess risk bound* holds:

$$\forall \delta \in (0, 1) \quad \Pr\left\{ \mathcal{L}(\hat{h}^*) \leq \inf_{h \in \mathcal{H}} \mathcal{L}(h) + \mathcal{C}'(n, \mathcal{H}, \delta) \right\} \geq 1 - \delta, \tag{2}$$

where $\mathcal{C}'(n, \mathcal{H}, \delta)$ is another generalization term as above.

**Multi-objective learning.** In multi-objective learning, multiple target objectives, $\mathcal{L}_1, \ldots, \mathcal{L}_N$, characterize different properties of interest of the hypotheses. Estimating them from a (single) dataset yields empirical objectives, $\widehat{\mathcal{L}}_1, \ldots, \widehat{\mathcal{L}}_N$. In contrast to the single-objective situation where the objective function is almost always related to a measure of prediction quality, the multi-objective setting provides a principled framework for expressing also other relevant quantities of a machine learning model, such as *efficiency*, *robustness*, or *fairness*. Consequently, we allow the objectives to also have other forms than just expected values over per-sample loss functions, and their empirical estimates are not restricted to per-sample averages. As discussed in Section 2.1, the multi-objective setting does not induce a total ordering of the hypotheses. Consequently, *a priori* there will be no overall *best* hypothesis anymore. Instead, there there are two sets of Pareto-optimal hypotheses:

**Definition 3.** a) A hypothesis $h \in \mathcal{H}$ is called *empirically Pareto-optimal* if it is Pareto-optimal with respect to the multi-objective optimization problem of minimizing $\widehat{\mathcal{L}}_1(h), \ldots, \widehat{\mathcal{L}}_N(h)$ (with are computed from some training set $S$). The set of all such hypotheses we call the *empirically Pareto-optimal set.*

b) A hypothesis $h \in \mathcal{H}$ is called *(truly) Pareto-optimal* if it is Pareto-optimal with respect to the multi-objective optimization problem of minimizing $\mathcal{L}_1(h), \ldots, \mathcal{L}_N(h)$. The set of all such hypotheses we call the *(truly) Pareto-optimal set.*

Analogously to single-objective learning, we are most interested in finding truly optimal hypotheses (here, e.g., the truly Pareto-optimal set), as these can be expected to work well on future data. However, we can only compute solutions to the empirical problem (the empirically Pareto-optimal set). If solutions to the latter problem approximate the former it is called *multi-objective generalization.*

In recent years, multi-objective learning has received increasing attention in the machine learning community, and a number of algorithms have been proposed for it. In their easiest form, one simply picks a scalarization method and solves the resulting single-objective optimization problem with fixed scalarization weights or one optimizes over those as well (Cortes et al., 2020; Deist et al., 2021; Fliege & Svaiter, 2000). Alternatively, one can search for hypotheses along the complete (empirically) Pareto-front, using, e.g., ensemble techniques (Liu & Kadirkamanathan, 1995; Van Veldhuizen & Lamont, 1998), model conditioning (Ruchte & Grabocka, 2021), or hypernetworks (Navon et al., 2021).

Given the long tradition and algorithmic diversity, one could expect *multi-objective statistical learning theory* also to be a rich field that provides precise quantifications of the relations between true and empirical objective (generalization bounds), as well as relation between the empirical and true Pareto-optimal sets (excess bounds). Surprisingly, this is not the case, and hardly any such results exist in the literature.

## 3 Related work

Solving problems with multiple objectives has a long tradition in artificial intelligence (Aziz et al., 2016; Deb, 2001; Rahwan & Larson, 2008; Zhou et al., 2011), game theory (Fudenberg & Tirole, 1991; Pardalos et al., 2008), and economics (Hochman & Rodgers, 1969; Keeney et al., 1993). Since the 1990s it has also attracted attention from the machine learning community, e.g. Fieldsend & Singh (2005); Goldberg (1989); Jin (2006). Existing works predominantly study the problem from an algorithmic perspective, in particular proposing and analyzing new optimization techniques. Mirroring the corresponding developments in multi-objective optimization, this includes methods for efficiently finding individual Pareto-optimal solutions, e.g. Cortes et al. (2020); Van Moffaert & Nowé (2014); Ye et al. (2021), as well as exploring the complete Pareto front (Jin & Sendhoff, 2008; Navon et al., 2021; Przybylski & Gandibleux, 2017; Ruchte & Grabocka, 2021; Vamplew et al., 2011; Van Moffaert & Nowé, 2014; Zhu & Jin, 2019). Works in both directions implicitly assume that better results of the empirical learning task should translate to better results on future data. So far, this *generalization* aspect was studied only empirically. Theoretical results rather focused on the optimization aspect, e.g. studying *computational complexity* (Teytaud, 2007; Wang & Sandholm, 2003) or *convergence rates* (Stark & Spall, 2003), but not statistical generalization. A notable exception is Cortes et al. (2020), which we discuss in detail in the Section 5.3.

## 4 Main results

In this section we formally state and discuss our main results: generalization and excess bounds for scalarizations and for Pareto-fronts. For maximal generality, we formulate the results on the generic level introduced in Section 2. We will discuss instantiations that either improve over related existing work or provide new insights in Sections 5 and we provide a high-level overview of potential additional applications in Section 6.

**Assumptions.** Because the multi-objective setting strictly generalizes the single-objective one, multi-objective generalization is not possible unless at least single-objective generalization holds. Therefore, for all our results we adopt the following assumption.

*Assumption A. — For each objective individually a generalization bound of the form* (1) *holds.*

Note that Assumption A is technically easy to fulfill, at least for bounded objectives, by setting the required generalization terms, $\mathcal{C}_i(n, \mathcal{H}, \delta)$ for $i \in [N]$, to large enough constants. Our results do hold for such a choice, but their interpretation would mostly not be very interesting. Therefore, whenever we want to interpret results in the light of their approximation quality, we additionally make the following assumption.

*Assumption B. — For each $i \in [N]$ and for each $\delta \in (0, 1)$, it holds that $\mathcal{C}_i(n, \mathcal{H}, \delta) \overset{n \to \infty}{\to} 0$.*

As we detail in Section 6, Assumption A and Assumption B are fulfilled for many quantities of interest related to the *accuracy*, *fairness*, *robustness* or *efficiency* of machine learning systems. Noteworthy special cases are objectives that are data-independent functions of only the hypothesis, for example, regularization terms. We say that such objectives *generalize trivially*, because they fulfill $\mathcal{L}(h) = \widehat{\mathcal{L}}(h)$ for all datasets and all $h \in \mathcal{H}$, and therefore generalization bounds of the form (1) hold for them trivially with 0 as generalization term.

### 4.1 Multi-objective generalization

Our first result states that if generalization bounds hold individually for each objective, then they hold also jointly in the multi-objective setting, where the empirical objectives are computed from a single dataset, at only a minor loss of confidence.

**Lemma 1** (Multi-Objective Generalization Bound). *Let $N_{nt}$ be the number of non-trivial objectives. Let $S$ be a random dataset of size $n$. For each $i \in [N]$, let $\widehat{\mathcal{L}}_i$ be an empirical estimate of $\mathcal{L}_i$ based on a subset $S_i \subset S$ of size $n_i$. Then it holds with probability at least $1 - \delta$,*

$$\forall i \in [N], \ \forall h \in \mathcal{H} : |\mathcal{L}_i(h) - \widehat{\mathcal{L}}_i(h)| \leq \mathcal{C}_i(n_i, \mathcal{H}, \delta/N_{nt}). \tag{3}$$

Lemma 1 is in fact a straight-forward consequence of Assumption A, requiring only a union-bound argument as proof. We state it explicitly nevertheless because it has not appeared in this form in the literature so far.

### 4.2 Generalization and excess bounds for scalarizations

A common way for learning in a multi-objective setting is by performing single-objective learning for one or multiple scalarizations. To keep the notation concise, for any scalarization $\mathcal{U} : \mathbb{R}^N_+ \to \mathbb{R}_+$ and $h \in \mathcal{H}$, we abbreviate $\mathcal{L}_{\mathcal{U}}(h) := \mathcal{U}(\mathcal{L}_1(h), \ldots, \mathcal{L}_N(h))$, $\widehat{\mathcal{L}}_{\mathcal{U}}(h) := \mathcal{U}(\widehat{\mathcal{L}}_1(h), \ldots, \widehat{\mathcal{L}}_N(h))$.

**Theorem 2** (Generalization and Excess Bounds for Scalarizations). *Assume the same setting as for Lemma 1. Let $\mathfrak{U} = \{\mathcal{U} : \mathbb{R}^N \to \mathbb{R}_+\}$ be a set of scalarizations, each of which is $L_{\mathcal{U}}$-Lipschitz continuous with respect to some monotonic norm $\|\cdot\|_{\mathcal{U}}$. Then, for all $\delta > 0$ the following two statements hold with probability at least $1 - \delta$.*

*a) For all $\mathcal{U} \in \mathfrak{U}$ and $h \in \mathcal{H}$:*

$$\left|\mathcal{L}_{\mathcal{U}}(h) - \widehat{\mathcal{L}}_{\mathcal{U}}(h)\right| \leq L_{\mathcal{U}} \left\|\left(\mathcal{C}_1(n_1, \mathcal{H}, \delta/N_{nt}), \ldots, \mathcal{C}_N(n_N, \mathcal{H}, \delta/N_{nt})\right)\right\|_{\mathcal{U}}. \tag{4}$$

*b) For all $\mathcal{U} \in \mathfrak{U}$, for all $\hat{h}^*_{\mathcal{U}} \in \arg\min_{h \in \mathcal{H}} \widehat{\mathcal{L}}_{\mathcal{U}}(h)$, and for all $h \in \mathcal{H}$:*

$$\mathcal{L}_{\mathcal{U}}(\hat{h}^*_{\mathcal{U}}) \leq \mathcal{L}_{\mathcal{U}}(h) + 2L_{\mathcal{U}} \left\|\left(\mathcal{C}_1(n_1, \mathcal{H}, \delta/N_{nt}), \ldots, \mathcal{C}_N(n_N, \mathcal{H}, \delta/N_{nt})\right)\right\|_{\mathcal{U}}. \tag{5}$$

**Proof sketch.** We provide the main arguments of the proofs here. The complete proofs are provided in Appendix A. a) The Lipschitz property implies that the difference of scalarized objectives is upper bounded by the norm of the differences in objective values. By the norm's monotonicity and Lemma 1, this is again bounded by the norm of the generalization terms. b) from $\widehat{\mathcal{L}}_{\mathcal{U}}(\hat{h}^*_{\mathcal{U}}) \leq \widehat{\mathcal{L}}_{\mathcal{U}}(h)$ it follows that $\mathcal{L}_{\mathcal{U}}(\hat{h}^*_{\mathcal{U}}) - \mathcal{L}_{\mathcal{U}}(h) \leq \mathcal{L}_{\mathcal{U}}(\hat{h}^*_{\mathcal{U}}) - \widehat{\mathcal{L}}_{\mathcal{U}}(\hat{h}^*_{\mathcal{U}}) + \widehat{\mathcal{L}}_{\mathcal{U}}(h) - \mathcal{L}_{\mathcal{U}}(h)$. Using a) we can bound the difference between the first two terms as well as the difference between the last two terms on the right hand side each by the norm of the generalization terms.

**Discussion.** Theorem 2 establishes *generalization* and *excess bounds* for the situation of scalarization-based multi-objective learning. Their relevance lies not only in the inequalities (4) and (5) themselves, which have

the standard single-objective form, but also in the fact that these hold *uniformly* over all scalarizations $\mathcal{U} \in \mathfrak{U}$. This implies that one can solve an arbitrary number of scalarized problems without suffering a loss of confidence in the theoretical guarantees. That is in contrast to other situations of repeated learning, e.g. hyperparameter-tuning on a validation set, where the statistical guarantees deteriorate with the number of hypotheses considered, because of the *multiple hypothesis testing* phenomenon (Shalev-Shwartz & Ben-David, 2014, Chapter 11). Despite its simplicity, the theorem improves over prior work, Cortes et al. (2020), which proved guarantees that depend on the size of $\mathfrak{U}$. For a more detailed discussion see Section 5.3.

### 4.3 Pareto excess bounds

We now provide a formal analysis of the relation between the set of Pareto-optimal hypotheses and the set of empirically Pareto-optimal hypotheses. First, we show that any two elements of the two Pareto-optimal sets fulfill an excess-type inequality with respect to at least some of the objectives.

**Theorem 3.** *Assume the same situation as for Lemma 1. Then, for any $\delta > 0$, it holds with probability at least $1 - \delta$: for all Pareto-optimal $h^* \in \mathcal{H}$ and empirically Pareto-optimal $\hat{h}^* \in \mathcal{H}$ there exists a non-empty subset $I \subset [N]$, such that*

$$\forall i \in I: \quad \mathcal{L}_i(\hat{h}^*) \leq \mathcal{L}_i(h^*) + 2\mathcal{C}_i(n_i, \mathcal{H}, \delta/N_{nt}). \tag{6}$$

**Proof sketch.** The proof works by contradiction: assume that a pair $(h^*, \hat{h}*)$ exists such that for no index set inequality (6) would hold. Then, using Lemma 1, one could show that $h^*$ strongly dominates $\hat{h}^*$ with respect to the empirical objectives, which is a contradiction to the optimality of $\hat{h}^*$. For the formal steps, see Appendix A. Like the sketch, the formal proof does not actually make use of the optimality of $h^*$. This implies that Theorem 3 holds in fact for all $h \in \mathcal{H}$, making it even more apparent that excess bound with respect to individual objectives are of limited use for studying multi-objective generalization.

For multi-objective learning the most relevant question is if there is an analog of Theorem 3 for the case of $I = [N]$, i.e. if by finding the empirical Pareto-curve one also approximately recovers the true Pareto-curve with respect to *all* objectives. This is formalized in the following theorem.

**Theorem 4** (Pareto Excess Bound). *Assume the same setting as for Lemma 1. Then, for any $\delta > 0$, it holds with probability at least $1 - \delta$.*

*a) For all Pareto-optimal $h^* \in \mathcal{H}$ there exists an empirically Pareto-optimal $\hat{h}^* \in \mathcal{H}$ with*

$$\forall i \in [N]: \quad \mathcal{L}_i(\hat{h}^*) \leq \mathcal{L}_i(h^*) + 2\mathcal{C}_i(n_i, \mathcal{H}, \delta/N_{nt}). \tag{7}$$

*b) Assume that the Pareto-front is* ray complete, *i.e. for all $R \in \{(r_1, \ldots, r_N) : r_i > 0 \text{ for } i \in [N]\}$, there exists an $h \in \mathcal{P}$ with $\left(\mathcal{L}_1(h), \ldots, \mathcal{L}_N(h)\right) \propto R$. Then, for all empirically Pareto-optimal $\hat{h}^* \in \mathcal{H}$, there exists a Pareto-optimal $h^* \in \mathcal{H}$ with*

$$\forall i \in [N]: \quad \mathcal{L}_i(\hat{h}^*) \leq \mathcal{L}_i(h^*) + 2\mathcal{C}_i(n_i, \mathcal{H}, \delta/N_{nt}). \tag{8}$$

**Proof sketch.** To prove part a), we make use of the fact that $h^* \in \mathcal{H}$ is dominated with respect to the empirical objectives by some empirically Pareto-optimal $h^* \in \mathcal{H}$, i.e. $\widehat{\mathcal{L}}_i(\hat{h}^*) \leq \widehat{\mathcal{L}}_i(h^*)$ for all $i \in [N]$. Statement (7) follow by applying Lemma 1 to both sides of this inequality and rearranging terms.

The main insight for proving part b) is that $\hat{h}^* \in \arg\min_{h \in \mathcal{H}} \widehat{\mathcal{L}}_{\mathcal{U}}(h)$ for the Chebyshev scalarization $\mathcal{U}(x_1, \ldots, x_N) = \max_{j \in [N]} w_j x_j$ with weights $w_j = \frac{1}{\widehat{\mathcal{L}}_j(\hat{h}^*)}$ for $j \in [N]$, as long as $\widehat{\mathcal{L}}_i(\hat{h}^*) > 0$ for all $i \in [N]$. With $h^* \in \arg\min_{h \in \mathcal{H}} \mathcal{L}_{\mathcal{U}}(h)$ it follows from Theorem 2 that $\max_{j \in [N]} w_j \mathcal{L}_j(\hat{h}^*) \leq \max_{j \in [N]} w_j \mathcal{L}_j(h^*) + 2\max_{j \in [N]} w_j \mathcal{C}_j(n_j, \mathcal{H}, \delta/N_{nt})$. For any $i \in [N]$, it holds that $w_i \mathcal{L}_i(\hat{h}^*) \leq \max_{j \in [N]} w_j \mathcal{L}_j(\hat{h}^*)$, and the assumption of ray completeness ensures that, $w_i \mathcal{L}_i(\hat{h}^*) = \max_{j \in [N]} w_j \mathcal{L}_j(\hat{h}^*)$. In combination, one obtains the same statement as (8), except with a potentially weaker generalization term $\frac{2}{w_i} \max_{j \in [N]} w_j \mathcal{C}_j(n_j, \mathcal{H}, \delta/N_{nt})$. To obtain the desired result, one creates additively shifted objective functions that result in a learning setting

equivalent to the original one, but in which all terms $w_j \mathcal{C}_j(n_j, \mathcal{H}, \delta/N_{\mathrm{nt}})$ for $j \in [N]$ are identical, such that, in particular, $\max_{j \in [N]} w_j \mathcal{C}_j(n_j, \mathcal{H}, \delta/N_{\mathrm{nt}}) = w_i \mathcal{C}_i(n_i, \mathcal{H}, \delta/N_{\mathrm{nt}})$. For complete proofs, see Appendix A.

**Discussion.** The theorems in this section clarify the relation between the true Pareto-optimal set and its empirical counterpart. When looking a single objective at a time, the relation is nearly trivial: Theorem 3 establishes that any empirically Pareto-optimal hypothesis is not much worse than any truly Pareto-optimal hypothesis with respect to *at least one* of the objectives.

More interesting is the situation when studying all objectives simultaneously. Theorem 4 provides a multi-objective analog of the classical *empirical risk minimization principle* (Vapnik, 2013). Solving the empirical multi-objective learning problem makes sense as a learning strategy, because for every truly Pareto-optimal hypothesis there is an empirically Pareto-optimal one that has not much larger (true) objective values, jointly across all of the objectives. Reversely, every empirically Pareto-optimal hypothesis is not substantially worse than some Pareto-optimal one, if we make an additional assumption on the geometry of the Pareto-front.

The employed *ray completeness* assumption is quite restrictive and we do not expect it to be fulfilled in most real-world situations. For example, it is violated already whenever one of the objectives is bounded from below by a constant bigger than 0.

In the two-objectives situation, ray completeness does hold if the Pareto-front is a continuous curve between some point on the $\mathcal{L}_1$-coordinate axis and some point on the $\mathcal{L}_2$-coordinate axis, excluding the origin. An example where such a situation can happen is a classification task in the realizable setting with *classification error* and (suitably defined) *computational cost* as objectives. For a sufficently rich hypothesis set, the smallest achievable error will be a continuous and mononotically decreasing function of the specified computational budget. Consequenty, the Pareto-front will be a continuous curve between a point $(a, 0)$, where $a$ is the classification error of the classifier with minimal budget, and a point $(0, b)$, where $b$ is the smallest computational cost for a classifier achieving minimal classification error. Note that realizability is necessary. Otherwise, the curve would still be monotonic, but the second point in the above construction would not lie on the $\mathcal{L}_2$-coordinate axis. Consequently, ray completeness would not be fulfilled.

While sufficient, ray completeness is certainly not a necessary condition. For example, if both the real objective and the empirical objective are bounded away from zero by the same constant, substracting this constant from both objectives could yield a situation that is equivalent in terms of multi-objective learning, but in which ray completeness might be fulfilled. Furthermore, from the Theorem's proof one can see that a weaker condition would suffice, namely that every ray through the empirical Pareto-front front also intersects the actual Pareto front. This formulation would complicate the condition, though, as it introduces a dependence on the dataset and would nevertheless still not be a mathematically necessary condition. Therefore, we leave the task of identiying a condition that is necessary as well as sufficient to future work.

The following theorem shows that some additional assumption is required for Theorem 4 to hold.

**Theorem 5.** *Let $N \geq 2$. Then, for any $C > 0$ there exist a learning problem that fulfills Assumptions A and B with $\mathcal{C}_i(n_i, \mathcal{H}, \delta) = 0$ for $i \in [N-1]$, but for which with probability at least $\frac{1}{2}$ there exists an empirically Pareto-optimal $\hat{h}^* \in \mathcal{H}$, such that for all Pareto-optimal $h^* \in \mathcal{H}$, it holds*

$$\forall i \in [N-1]: \quad \mathcal{L}_i(\hat{h}^*) > \mathcal{L}_i(h^*) + C. \tag{9}$$

**Proof.** We prove the theorem by constructing a concrete counterexample that exploits the classic *overfitting* (or *bias-variance trade-off*) phenomenon of single-objective supervised learning (Vapnik, 2013).

First, we look at the case $N = 2$. Let $\mathcal{Z} = \mathcal{X} \times \mathcal{Y}$ with $\mathcal{X} = [0, 1]$ and $\mathcal{Y} = \{0, 1\}$, $p(x)$ be the uniform distribution and $p(y|x) = \frac{1}{2}$. Let $\mathcal{H} = \{h : \mathcal{X} \to \mathcal{Y}\}$ be the set of piecewise-constant functions that consist of at most $K$ segments. We choose the number of jumps as $\mathcal{L}_1$ and the 0/1-loss as $\mathcal{L}_2$. Then, Assumption A and Assumption B are fulfilled: $\mathcal{H}$ is known to have VC-dimension $2k$ (Shalev-Shwartz & Ben-David, 2014), so a classical generalization bound holds for $\mathcal{L}_2$. $\mathcal{L}_1$ even generalizes trivially.

We observe that every hypothesis in $h \in \mathcal{H}$ fulfills $\mathcal{L}_2(h) = \frac{1}{2}$. Consequently, the two Pareto-optimal solutions, $h$, are the constant classifiers, which fulfill $\mathcal{L}_2(h) = \frac{1}{2}$ and $\mathcal{L}_1(h) = 0$.

Empirically, however, for sufficiently many points, with high probability, the empirical loss $\widehat{\mathcal{L}}_2(h)$ will be strictly monotonically decreasing with respect to $\widehat{\mathcal{L}}_1(h)$, as more segments allow to better fit the training data. Consequently, the set of empirically Pareto-optimal solutions will contain elements with $\widehat{\mathcal{L}}_1(h) = k$ for any $k \in [K]$, i.e. arbitrarily far from all solutions in the truly Pareto-optimal set. Figure 2 shows a visualization of this situation.

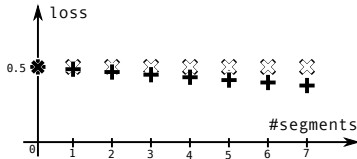

Figure 2: Illustration of the counterexample proving Theorem 5 for $N = 2$

For larger $N$, we use the analogous construction in $\mathbb{R}^{N-1}$. Hypotheses have at most $K$ jumps in each coordinate dimension. Objectives 1 to $N-1$ are the number of jump per coordinate; objective $N$ is the classification 0/1-loss.

**Discussion.** Theorem 5 establishes that the asymmetry between the statements a) and b) of Theorem 4 is an intrinsic property of the multi-objective setting, not a limitation of our proof techniques. There can indeed be hypotheses in the empirically Pareto-optimal set that are not in an excess relation with any hypothesis in the truly Pareto-optimal set. Note that despite the fact that multi-objective learning includes single-objective learning as a special case, there is no contradiction to the classical symmetric result. For $N = 1$, the fact that $I \subset [N]$ is non-empty in Theorem 3 makes its statement identical to Theorem 4 b) without the additional assumption. Theorem 5 holds only for $N \geq 2$.

### 4.4 Summary

In combination, the results of this section establish a detailed picture of similarities and differences between the generalization properties of single-objective and multi-objective learning. In particular, it highlights a fundamental difference between single-objective learning and Pareto-based multi-objective learning: in the single-objective setting, empirical risk minimization is a good learning strategy, because with growing data, the every minimizer of the empirical risk also has close to optimal true risk. In the multi-objective setting, scalarized learning has the same properties, but the resulting guarantees hold only with respect to the scalarization of the objectives, not each of them individually. Joint statements across all objectives hold as well, thereby justifying Pareto-based learning. However, without additional assumptions excess guarantees can only be given for a subset of the empirical solutions.

## 5 Applications

Our results of the previous section provide new tools for analyzing learning tasks in which multiple, potentially competing, quantities are of simultaneous interest, such as *fairness*, *robustness*, *efficiency* and *interpretability*. So far, the generalization properties of these quantities have been studied either not at all, or only with task-specific tools. Similarly, we expect that *multi-task*, *multi-label* and *meta-learning*, as well as *AutoML* will potentially be able to benefit from the multi-objective view.

In the rest of this section, we sketch three exemplary applications. In Section 5.1 we show how our results on empirical versus true Pareto-optimality can provide new insights for a well-known method. In Section 5.2 we demonstrate how our results on scalarized learning provide a simple and flexible way for constructing new single-objective generalization bounds. In Section 5.3, we improve an existing generalization bound for the multi-objective setting, which thanks to our results of Section 4 requires only a few lines of proof. For a more high-level discussion of other application scenarios, see Section 6.

### 5.1 Simultaneous feature selection and regression

The classical LASSO method (Tibshirani, 1996) learns a linear regression function by solving the following regularized risk minimization problem

$$\min_{\beta \in \mathbb{R}^d} \quad \frac{1}{n} \sum_{i=1}^{n} (y_i - \beta^\top x_i)^2 + \lambda \|\beta\|_{L^1}. \tag{10}$$

Here $\{(x_1, y_1), \ldots, (x_n, y_n)\} \subset \mathbb{R}^d \times \mathbb{R}$ is a given training set and $\lambda \in \mathbb{R}^+$ is a free parameter that trades off the data fidelity on the training set with the sparsity of the coefficient vector $\beta$ at its optimum. LASSO is particularly popular because it not only learns a regression function but also performs feature selection and therefore can give rise to more interpretable regression models than other regression techniques. The set of all solutions obtained by minimizing (10) for different values of $\lambda$ is called the *solution path*. A number of efficient techniques for computing it have been developed (Efron et al., 2004; Gaines et al., 2018; Osborne et al., 2000).

We can interpret the LASSO problem equivalently as the linear scalarization of a two-objective learning problem. The first objective is the expected squared loss $\mathcal{L}_1(\beta) = \mathbb{E}_{(x,y)}(y - \beta^\top x)^2$, which has $\widehat{\mathcal{L}}_1(\beta) = \frac{1}{n} \sum_{i=1}^n (y_i - \beta^\top x_i)^2$ as empirical counterpart. When the data and coefficient vector come from bounded domains, standard generalization bounds are known to hold that relate $\mathcal{L}_1$ and $\widehat{\mathcal{L}}_1$, see e.g. Mohri et al. (2018, Theorem 11.11). The second objective is the regularizer, $\mathcal{L}_2(\beta) = \widehat{\mathcal{L}}_2(\beta) = \|\beta\|_{L^1}$, which generalizes trivially.

The (single-objective) generalization properties of LASSO's squared loss term are well understood. The multi-objective view, however, adds insight into its relation with the regularizer, which reflects the sparsity and thereby the interpretability of the solutions. First, we observe that the underlying optimization problem is convex, so each empirically Pareto-optimal solution can be recovered by solving (10) for some value of $\lambda$. Therefore, existing *solution path* methods can readily be used to identify the empirical Pareto front.

Theorem 4 a) now ensures that each truly Pareto-optimal solution can be approximately recovered this way. This means, we can be sure that no solutions exist that are substantially sparser at equal accuracy or more accurate at identical sparsity with respect to the true objectives than some in the solution path.

However, Theorem 5 reminds us that not all solutions found by solving (10) will necessarily be close to truly Pareto-optimal. In particular, this means, while each individual element of the solution path will have optimal sparsity for the empirical accuracy it achieves, its sparsity might be far from optimal compared to other solutions of similar true accuracy. Consequently, if optimal sparsity is important for the task at hand, the solutions on the regularization path should be further evaluated, e.g. using validation data.

The latter comment does not apply if the true underlying regression task is actually linear, such that a coefficient vector, $\beta$, exists with vanishing objective, $\mathcal{L}_1(\beta) = 0$. Because the same property holds for $\mathcal{L}_2$ (trivially achieved by $\beta = 0$), and the regulariation path is a connected set (Tibshirani, 2013), the condition of *ray completeness* would be fulfilled. Theorem 4 b) then guarantees that in fact all empirically Pareto-optimal solutions are also approximately truly Pareto-optimal.

## 5.2 Tilted empirical risk minimization

Tilted empirical risk minimization (TERM) (Li et al., 2021) has recently been proposed as a widely applicable technique for making learning problem more *robust* or more *fair*. In its group-based form, TERM consists of minimizing an exponentially weighted risk functional

$$J_t(f) = \frac{1}{t} \log \Big( \frac{1}{N} \sum_{i=1}^N e^{tR_i(f)} \Big) \quad \text{with} \quad R_i(f) = \frac{1}{n} \sum_{(x,y) \in S_i} \ell(y, f(x)) \tag{11}$$

for a loss function $\ell$ and training data given as $N$ potentially overlapping groups, $S_1, \ldots, S_N$. For simplicity of explosition we assume all groups to be of identical sizes, $n$. The tilt parameter $t \in \mathbb{R} \setminus \{0\}$ determines whether the effect of TERM is to provide *robustness* against outlier groups ($t < 0$), or to encourage *fairness* between all groups ($t > 0$). For $t \to \infty$ and $t \to -\infty$, TERM converges to $\max_i R_i(f)$ and $\min_i R_i(f)$, respectively.

Taking a multi-objective perspective, $J_t$ in (11) can be seen as a parametrized family of scalarizations of empirical objectives $R_1, \ldots, R_N$. Each $J_t$ is 1-Lipschitz with respect to the $L^\infty$ norm. Assuming that generalization bounds hold for each individual group, then Theorem 2 guarantees that a generalization bound of the same structure holds also for $J_t$, simultaneously across all values of $t$.

*Hierarchical TERM (hTERM)* (Li et al., 2021) uses the same exponentially weighted functional $J_t$, but the per-group terms $R_1, \ldots, R_K$ that it combines are not averages across samples as in (11), but TERM-losses

themselves with individual tilt parameters $\tau_1, \ldots, \tau_K$. From our previous analysis, we know generalization bounds for each of those. Consequently, by the same construction as above, we readily obtain a generalization bound for hTERM, which in fact holds uniformly across all combinations of tilt parameters.

### 5.3 Agnostic learning with multiple objectives

Cortes et al. (2020) studies the generalization properties of multi-objective learning for the case of a special scalarization obtained by minimizing over convex combinations. To allow for an easier comparison, we state their result in our notation.[2]

**Theorem 6** (Theorem 3 in Cortes et al. (2020))**.** *Let $\mathcal{H}$ be a hypothesis set for a supervised learning problem with input set $\mathcal{X}$ and output set $\mathcal{Y}$, that fulfills $\|(h(x'), y') - (h(x), y)\| \leq D$ for some constant $D > 0$ and for all $(x, y), (x', y') \in \mathcal{X} \times \mathcal{Y}$. Let $\ell_i : \mathcal{Y} \times \mathcal{Y} \to \mathbb{R}_+$ for $i \in [N]$ be loss functions that are $M_i$-Lipschitz and upper-bounded by $M$. Set $\mathcal{L}_i(h) = \mathbb{E}_{(x,y)}[\ell(y, h(x)]$ and $\widehat{\mathcal{L}}_i(h) = \frac{1}{n}\sum_{i=1}^n \ell(y_i, h(x_i))$ for a dataset $S \overset{i.i.d.}{\sim} p(x,y)$ of size $n$. For a set of scalarization weights, $W \subset \Delta_N$, let $\mathcal{L}_W(h) = \max_{w \in W} \sum_{i=1}^N w_i \mathcal{L}_i(h)$ and $\widehat{\mathcal{L}}_W(h) = \max_{w \in W} \sum_{i=1}^N w_i \widehat{\mathcal{L}}_i(h)$. Assume that $\sum_{i=1}^N w_i M_i \leq \beta$ for all $w = (w_1, \ldots, w_N) \in W$.*

*Then, for any $\epsilon > 0$ and $\delta \in (0,1)$, with probability at least $1 - \delta$, the following inequality holds for all $h \in \mathcal{H}$:*

$$\mathcal{L}_W(h) \leq \widehat{\mathcal{L}}_W(h) + 2\beta\hat{\mathfrak{R}}_S(\mathcal{H}) + M\epsilon + 3\beta D\sqrt{\frac{1}{2n}\log\left[\frac{2|W_\epsilon|}{\delta}\right]}, \tag{12}$$

*where $\hat{\mathfrak{R}}_S(\mathcal{H})$ is the* empirical Rademacher complexity *of the hypothesis class $\mathcal{H}$ with respect to $S$, and $|W_\epsilon|$ is the size of a minimal $\epsilon$-cover of $W$.*

One can see that inequality (12) precisely matches the form of our excess bound in Theorem 2 for a specific scalarization. Indeed, we can derive an analogous theorem using our results of Section 4.

**Theorem 7.** *Make the same assumptions as in Theorem 6. Then, for any $\delta \in (0,1)$, with probability at least $1 - \delta$ the following inequality holds for all $h \in \mathcal{H}$:*

$$\mathcal{L}_W(h) \leq \widehat{\mathcal{L}}_W(h) + 2\beta\hat{\mathfrak{R}}_S(\mathcal{H}) + 3\beta D\sqrt{\frac{\log\frac{2N}{\delta}}{2n}}. \tag{13}$$

**Proof.** The Lipschitz assumptions imply standard Rademacher-based generalization bounds for each objective individually (Mohri et al., 2018, Theorem 11.3): with probability at least $1 - \delta$ it holds for all $h \in \mathcal{H}$:

$$\mathcal{L}_i(h) \leq \widehat{\mathcal{L}}_i(h) + 2M_i\hat{\mathfrak{R}}_S(\mathcal{H}) + 3M_i D\sqrt{\frac{\log\frac{2}{\delta}}{2n}}. \tag{14}$$

We now apply Theorem 2 for the family of linear scalarizations, $\mathcal{U}_w(x_1, \ldots, x_N) = \sum_{i=1}^N w_i x_i$ with $w \in W$, and we insert that $w_i M_i \leq \beta$. Finally, taking the maximum over $w$ on both sides of the inequality yields (13).

Because of the power of the introduced multiobjective framework, the proof of Theorem 7 is much shorter than the original one of Theorem 6. Nevertheless, our result has a number of advantages. First, our bound is structully simpler. It holds without need for an $\epsilon$-parameter that additively enters the right hand side of (12), yet also influences the size of the right-most confidence term. Second, the right hand side of our bound is independent of the size of $W$, with the confidence term only depending on the number of objectives. As a consequence, our bound is substantially tighter, except for trivially small sets $W$. For example, for the common case of convex combinations, $W = \Delta_N$, the covering size $|W|_\epsilon$ is of order $(1/\epsilon)^{N-1}$. This makes the generalization term in (12) of order $\sqrt{N/n}$, indicating that to preserve confidence the amount of data has to grow linearly with the number of objectives considered. In contrast, the right hand side of our bound (13) is independent of the size of $W$ and its confidence term grows only logarithmically with respect to $N$. Finally, our proof is not only simpler than the original one but also more flexible: it readily extends to other generalization bounds rather than just Rademacher-based ones, and to other scalarization besides linear combinations.

---

[2]Our formulation also has slightly different constants in the generalization term, which we believe to be necessary based on the theorem's proof.

## 6 Further application scenarios

In Section 5 we highlighted some specific examples in which our proposed multi-objective generalization theory provides new insights into existing methods. In this section, we provide more high-level background and discuss additional quantities that we believe will or will not benefit from a multi-objective analysis.

**Fairness.** *Algorithmic (group) fairness* asks to create classifiers that are not only accurate but also do not discriminate against certain protected groups in their decisions. Formally, this property can be expressed by different *(un)fairness measures*, such as *demographic parity*, *equality of opportunity* or *equalized odds* (Barocas et al., 2019). Because accuracy and fairness can be in conflict with each other, fairness-aware learning is a prototypical candidate for multi-objective learning (Martinez et al., 2020; Wei & Niethammer, 2020; Kamani et al., 2021; Padh et al., 2021). This view also extends naturally to integration of multiple fairness measures (Liu & Vicente, 2020), which might be incompatible with each other (Kleinberg et al., 2016; Chouldechova, 2017; Berk et al., 2021). Generalization bounds for the empirical estimation of unfairness measures have been developed (Woodworth et al., 2017; Konstantinov & Lampert, 2022). Consequently, our results from Section 4 apply, yielding a unified understanding of the generalization properties of fairness-aware learning, e.g., regularization-based (Kamishima et al., 2011) constraint-based (Calders et al., 2009), or Pareto-based (Liu & Vicente, 2020; Navon et al., 2021). The multi-objective view also allows us to conjecture that methods that seek fair hypotheses by other means, such as pre-processing (Kamiran & Calders, 2012) or post-processing (Hardt et al., 2016a), might not reach (empirically) Pareto-optimal solutions. If generalization guarantees do actually hold for these, other ways for proving them would be required.

**Robustness.** It has been observed that deep network classifiers in continuous domains such as image classification are susceptible to *adversarial examples*, i.e. they are not robust against small perturbation of the input data. Two main research directions have emerged to overcome this limitation: *Adversarial training* (Madry et al., 2018) adds a robustness-enforcing loss term to the training problem. Generalization bounds for such terms have been derived, e.g. Yin et al. (2019). Consequently, multi-objective learning can be used in this setting with the guarantees and caveats discussed above. *Lipschitz-networks* (Cisse et al., 2017) restrict the hypothesis class to functions with a small Lipschitz constant, typically 1. Afterwards one solves a training problem that tries to enforce a large margin between the predicted class label and the runner-up. From the achieved margins one can infer how large an input perturbation the classifier can tolerate without changing its decision (Weng et al., 2018). We are not aware of existing theoretical studies of such *certified robustness* techniques. However, margin-based loss functions have a long tradition in machine learning, and a number of generalization bounds exist which are applicable in the described situation, such as Kuznetsov et al. (2015); Koltchinskii & Panchenko (2002).

**Efficiency.** Large machine learning models, in particular deep networks, often have high computational demands, not only at training but also at prediction time (Strubell et al., 2020; Menghani, 2021). Consequently, a number of techniques have been developed that aim at reducing the computational cost. *Parameter sparsification* (Hoefler et al., 2021) and *quantization* (Gholami et al., 2021) are widely used methods for reducing the number of operations required to evaluate a model. As data-independent properties they can readily be used as *trivially-generalizing* objectives in a multi-objective learning framework (Zhu & Jin, 2019). Alternatively, speedup can also be achieved by encouraging as many zero values as possible to occur as part of the internal computation steps of a deep network. Such *activation sparsity* (Kurtz et al., 2020) is a data-dependent quantity that can also be shown to generalize using standard techniques. Therefore it as well can be handled in a multi-objective way. *Adaptive computation* methods, such as *ensembles* (Schwing et al., 2011), *classifier cascades* (Viola & Jones, 2001) or *multi-exit architectures* (Huang et al., 2018; Teerapittayanon et al., 2016), evaluate different subsets of a larger model depending on the input sample. For suitable design choices, generalization bounds for the resulting computation time can be proven, and our results will apply.

**Multi-task and multi-label learning.** *Multi-task learning* has recently been put forward as a multi-objective task, where each task's loss is treated as a separate objective (Sener & Koltun, 2018; Lin et al., 2019; Ma et al., 2020; Mahapatra & Rajan, 2020). This setting is of a non-standard form, as each task typically has a dedicated training set. Nevertheless, our framework can handle this setting as well, making use of the property that we allow the empirical estimates of different objectives to be derived from different subsets of the available data. Pareto-based guarantees are particularly relevant then, because at prediction

time, for each sample one is interested in only one of the objectives, namely the one of the task to which this sample belongs. In the related problems of *multi-label learning* (Zhang & Zhou, 2013) and *extreme classification* (Varma, 2019), the goal is to predict multiple outputs (labels) for each sample. Each label has an associated classifier objective, and the losses are estimated either from the total dataset or from (typically overlapping) subsets (Shi et al., 2012). Again, our framework is flexible enough to handle this setting. At prediction time all labels are meant to be predicted, and the quality is typically judged by a task-dependent aggregate measure, making scalarization approaches of particular interest in this setting.

**Limitations.** Despite its generality, some multi-objective learning settings do not lend themselves to an analysis using our results. For example, in the *learning-to-rank* setting (Liu, 2009) solutions are typically judged by two measures: *precision* and *recall*. *A priori*, this makes it a promising setting for multi-objective analysis (Cao et al., 2020; Svore et al., 2011). Unfortunately, we are not aware of generalization bounds for the *precision* objective. Given that its value fluctuates heavily in the low-recall regime, it is in fact possible that Assumption B might not be fulfillable. Also in the context of ranking, two other common objectives are *true positive rate (TPR)* and *false positive rate (FPR)*, which together trace out the *receiver operating characteristic (ROC) curve*. TPRs and FPRs can summarized into a single value by the *area under the ROC curve (AUC)* (Hanley & McNeil, 1982), for which indeed generalization bounds have been derived (Agarwal et al., 2005). However, the AUC is not a scalarization in the sense of Section 2.1, so Theorem 2 does not apply to it. Finally, besides the uniform generalization bounds of Assumption A, other guarantees of generalization have been developed, e.g., based on PAC-Bayesian theory (Dziugaite & Roy, 2017; McAllester, 1999), or *algorithmic stability* (Bousquet & Elisseeff, 2002; Hardt et al., 2016b). We see no principled reasons why results similar to ours should not hold for such settings as well, but other techniques would be required that lie outside of the scope of this work.

## 7 Conclusion

In this work, we proved a number of foundational results for the generalization theory of multi-objective learning. In particular, we showed that generalization bounds for the individual objectives imply generalization and excess bounds for multi-objective learning using scalarizations. Our second main result is an analysis of the relation between the Pareto-optimal sets of the empirical and the true learning problem. This justifies the use of Pareto-based methods on empirical data to approximately find all truly Pareto-optimal solutions. However, there is a caveat that some of the solutions found might be close to Pareto-optimal ones only with respect to some of the objectives, not all of them.

We formulated our results on a high level of generality that applies not only to measures of per-sample prediction quality, for which generalization bounds were originally developed, but also many other quantities of interest for modern machine learning systems, such as *fairness*, *robustness*, and *efficiency*. While initial results for some of these specific domains exist, we expect that more and stronger guarantees will be possible by more refined objective-specific analyses.

On a technical level, we see two directions for potentially improving our results. First, it would be desirable to have an explicit rather than implicit relationship between Pareto-optimal hypotheses and their best empirically Pareto-optimal approximations. Theorem 4 does not provide this. Even though its proof contains an explicit procedure, it relies on uncomputable quantities, such as the true objective objective values. Second, given that Theorem 5 establishes that there can be empirically Pareto-optimal hypotheses that do not approximate any truly Pareto-optimal hypothesis with respect to all objectives, it would be desirable to have an algorithmic procedure for testing which hypotheses these are. We see these as interesting directions for future work.

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

## A   Appendix – proofs of the main results

**Proof of Theorem 2.** With probability at least $1 - \delta$ for the dataset $S$ the relations of Lemma 1 will hold. By studying only these cases, we again obtaining results that hold with probability at least $1 - \delta$.

For statement a), for any $\mathcal{U} \in \mathfrak{U}$ we obtain by the Lipschitz property of the scalarization and Lemma 1 that for all $h \in \mathcal{H}$:

$$|\mathcal{L}_{\mathcal{U}}(h) - \widehat{\mathcal{L}}_{\mathcal{U}}(h)| \le L_{\mathcal{U}} \|\big(\mathcal{L}_1(h) - \widehat{\mathcal{L}}_1(h), \ldots, \mathcal{L}_N(h) - \widehat{\mathcal{L}}_N(h)\big)\|_{\mathcal{U}} \tag{15}$$

$$\le L_{\mathcal{U}} \|\big(|\mathcal{L}_1(h) - \widehat{\mathcal{L}}_1(h)|, \ldots, |\mathcal{L}_N(h) - \widehat{\mathcal{L}}_N(h)|\big)\|_{\mathcal{U}} \tag{16}$$

$$\le L_{\mathcal{U}} \|\big(\mathcal{C}_1(n_1, \mathcal{H}, \delta/N_{\mathrm{nt}}), \ldots, \mathcal{C}_N(n_N, \mathcal{H}, \delta/N_{\mathrm{nt}})\big)\|_{\mathcal{U}} \tag{17}$$

where the last two inequalities hold because of the norms' monotonicity, i.e. the fact that it is non-decreasing under increases of the input vector components Bauer et al. (1961). In combination, this proves statement a).

Statement b) follows by arguments mirroring the proof of classic *excess risk bounds* (Mohri et al., 2018). Let $\hat{h}_{\mathcal{U}}^* \in \arg\min_{h \in \mathcal{H}} \widehat{\mathcal{L}}_{\mathcal{U}}(h)$. Then, it holds for arbitrary $h \in \mathcal{H}$ that

$$\mathcal{L}_{\mathcal{U}}(\hat{h}_{\mathcal{U}}^*) - \mathcal{L}_{\mathcal{U}}(h) \le \mathcal{L}_{\mathcal{U}}(\hat{h}_{\mathcal{U}}^*) - \widehat{\mathcal{L}}_{\mathcal{U}}(\hat{h}_{\mathcal{U}}^*) + \widehat{\mathcal{L}}_{\mathcal{U}}(h) - \mathcal{L}_{\mathcal{U}}(h) \tag{18}$$

$$\le 2\|\big(\mathcal{C}_1(n_1, \mathcal{H}, \delta/N_{\mathrm{nt}}), \ldots, \mathcal{C}_N(n_N, \mathcal{H}, \delta/N_{\mathrm{nt}})\big)\|_{\mathcal{U}} \tag{19}$$

where the first inequality holds because $\widehat{\mathcal{L}}_{\mathcal{U}}(\hat{h}_{\mathcal{U}}^*) \le \widehat{\mathcal{L}}_{\mathcal{U}}(h)$ by construction of $\hat{h}_{\mathcal{U}}^*$, and the second inequality one follows from applying (4) twice, once for $h$ and once for $\hat{h}_{\mathcal{U}}^*$. The statement of the theorem now follows by moving the term containing $h$ to the right hand side.

**Proof of Theorem 3.** We again only study the case in the inequalities of Lemma 1 are fulfilled, so the results we achieve hold with probability at least $1 - \delta$.

We prove the remaining part of the theorem by contradiction. The negation of the statement reads: *there exists an empirically Pareto-optimal hypothesis $\hat{h}^* \in \mathcal{H}$ and a hypothesis $h \in \mathcal{H}$ such that $\mathcal{L}_i(\hat{h}^*) - \mathcal{L}_i(h) > 2\mathcal{C}_i(n_i, \mathcal{H}, \delta/N_{nt})$ for all $i \in [N]$.*

For these $\hat{h}^*$ and $h$ it follows that for all $i \in [N]$:

$$\widehat{\mathcal{L}}_i(h) - \widehat{\mathcal{L}}_i(\hat{h}^*) \leq \mathcal{L}_i(h) - \mathcal{L}_i(\hat{h}^*) + 2\mathcal{C}_i(n_i, \mathcal{H}, \delta/N_{\text{nt}}) < 0. \tag{20}$$

For the first inequality we applied Lemma 1 twice, and the second inequality follows from the assumption. However, (20) establishes that $h$ empirically strongly dominates $\hat{h}^*$ which is a contradiction to the assumption that $\hat{h}^*$ was empirically Pareto-optimal.

**Proof of Theorem 4** We again only study these case in which the dataset fulfills the inequalities of Lemma 1, so the results we achieve holds with probability at least $1 - \delta$.

Statement a) is a consequence of Lemma 1 and the definition of (empirical) Pareto-optimality. Let $h^* \in \mathcal{H}$ be Pareto-optimal. If it is also empirically Pareto-optimal, inequality (7) holds trivially with $\hat{h}^* = h^*$. Otherwise, there exists an empirically Pareto-optimal $\hat{h}^*$ that dominates $h^*$ with respect to the empirical objectives, i.e. in particular $\widehat{\mathcal{L}}_i(\hat{h}^*) \leq \widehat{\mathcal{L}}_i(h^*)$ for all $i \in [N]$. From this, we obtain for all $i \in [N]$, analogously to the proof of Theorem 2b):

$$\mathcal{L}_i(\hat{h}^*) - \mathcal{L}_i(h^*) \leq \mathcal{L}_i(\hat{h}^*) - \widehat{\mathcal{L}}_i(\hat{h}^*) + \widehat{\mathcal{L}}_i(h^*) - \mathcal{L}_i(h^*) \leq 2\mathcal{C}_i(n_i, \mathcal{H}, \delta/N_{\text{nt}}). \tag{21}$$

Before proving statement b) we introduce *additively shifted objectives.* as the main tool.

**Definition 4.** For an objective $\mathcal{L}(h)$ with empirical estimate $\widehat{\mathcal{L}}(h)$ and a constant $K$, we call $\mathcal{L}^{+K}(h) = \mathcal{L}(h) + K$ and $\widehat{\mathcal{L}}^{+K}(h) = \widehat{\mathcal{L}}(h) + K$ their $K$-additively shifted variants.

Generalization and Pareto-optimality are unaffected by additive shifts.

**Lemma 8.** *a) For any constant $K$, if a generalization bound of the form* (1) *holds for an objective $\mathcal{L}$ and its empirical estimate $\widehat{\mathcal{L}}$, then a bound with identical generalization term also holds for $\mathcal{L}^{+K}$ and $\widehat{\mathcal{L}}^{+K}$. b) For any constants $K_1, \ldots, K_N$, a solution $h \in \mathcal{H}$ is Pareto-optimal for $\mathcal{L}_1, \ldots, \mathcal{L}_N$ if and only if it is Pareto-optimal for $\mathcal{L}_1^{+K_1}, \ldots, \mathcal{L}_1^{+K_N}$. The analogous relation holds for empirically Pareto-optimality.*

The proofs are elementary: for a) the additive terms cancel out in the generalization bound. For b) Pareto-optimality depends only on the the relative order of objective values, which is not affected by additive shifts.

**Lemma 9.** *Let $h^* \in \mathcal{H}$ be a Pareto-optimal solution with $\mathcal{L}_i(h) > 0$ for all $i \in [N]$. Then $h^*$ is a minimizer to the Chebyshev scalarization $\mathcal{U}_w^{(\infty)}(h) = \max_{i \in [N]} w_i \mathcal{L}_i(h)$ with weights $w_i = \frac{1}{\mathcal{L}_i(h^*)}$ for $i \in [N]$. Furthermore, for any other minimizer, $h^\dagger$, of the scalarization it holds that $\mathcal{L}_i(h^\dagger) = \mathcal{L}_i(h^*)$ for all $i \in [N]$. The analogous result holds for empirically Pareto-optimal hypotheses.*

*Proof.* We prove the lemma by contradiction. First, assume $h$ to be a hypothesis with strictly smaller value for the scalarization. By construction $w_i \mathcal{L}_i(h^*) = 1$ for all $i \in [N]$, therefore $w_i \mathcal{L}_i(h) < 1$ for all $i \in [N]$ must hold. This, however, would imply $\mathcal{L}_i(h) < \mathcal{L}_i(h^*)$ for all $i \in [N]$, which is impossible because $h^*$ is Pareto-optimal. For $h^\dagger$, we know $w_i \mathcal{L}_i(h^\dagger) \leq 1$ and therefore $\mathcal{L}_i(h^\dagger) \leq \mathcal{L}_i(h^*)$ for all $i \in [N]$. Because of $h^*$'s Pareto-optimality, none of these inequalities can be strict, which proves the statement. The same line of arguments holds in the empirical situation. $\square$

We now turn to the proof of Theorem 4 b). Let $\hat{h}^*$ be an empirically Pareto-optimal solution for $\mathcal{L}_1, \ldots, \mathcal{L}_N$. For a more concise notation, we abbreviate $c_i = \mathcal{C}_i(n, \mathcal{H}, \delta/N_{\text{nt}})$.

First, we consider the case where none of the objectives are trivially generalizing, i.e. $c_i > 0$ for all $i \in [N]$. By Lemma 8, we know that $h^*$ is also empirically Pareto-optimal for the shifted objectives $\widehat{\mathcal{L}}_1^{+K_1}, \ldots, \widehat{\mathcal{L}}_{N'}^{+K_{N'}}$ with

$$K_i := Cc_i - \widehat{\mathcal{L}}_i(h^*) \quad \text{for } C = 2 + \max_j [\frac{1}{c_j}(\widehat{\mathcal{L}}_j(h^*) - \min_h \widehat{\mathcal{L}}_j(h))] \tag{22}$$

An explicit calculation confirms that $\widehat{\mathcal{L}}_i^{+K_i}(h) \geq 2c_i > 0$, which by assumption implies $\mathcal{L}_i^{+K_i}(h) \geq c_i > 0$, for all $i \in [N]$. By Lemma 9 we know that $h^*$ is a minimizer of the Chebyshev scalarization with weights $w_i = \frac{1}{\mathcal{L}_i^{+K_i}(h^*)} = \frac{1}{Cc_i}$ for all $i \in [N']$. Let $h^*$ be a minimizer of the scalarization of the true objectives with same weights $w_i$. The assumption of *ray completeness* together with Lemma 9 implies $w_1 \mathcal{L}_1^{+K_1}(h^*) = \cdots = w_N \mathcal{L}_N^{+K_N}(h^*) = \max_{j \in [N]} w_j \mathcal{L}_N^{+K_j}(h^*)$. The Chebyshev scalarization is a weighted $L^{(\infty)}$-norm and 1-Lipschitz with respect to itself. Therefore, by Theorem 2:

$$\max_{j \in [N]} w_j \mathcal{L}_j^{+K_j}(\hat{h}^*) \leq \max_{j \in [N]} w_j \mathcal{L}_j^{+K_j}(h^*) + 2 \max_{j \in [N]} w_j c_j \tag{23}$$

Consequently, we obtain the component-wise inequalities:

$$\forall i \in [N] \qquad w_i \mathcal{L}_i^{+K_i}(\hat{h}^*) \leq w_i \mathcal{L}_i^{+K_i}(h^*) + 2 \max_{j \in [N]} w_j c_j \tag{24}$$

Now, inserting the definition $K_i$, subtracting $w_i K_i$ from both sides and dividing by $w_i$ we obtain

$$\forall i \in [N] \qquad \mathcal{L}_i(\hat{h}^*) \leq \mathcal{L}_i(h^*) + \frac{2}{w_i} \max_{j \in [N]} w_j c_j. \tag{25}$$

By construction, $w_j c_j = \frac{1}{C}$ for all $j \in [N]$. Therefore, $\frac{2}{w_i} \max_{j \in [N]} w_j c_j = \frac{2Cc_i}{C} = 2c_i$. Because $c_i = \mathcal{C}_i(n_i, \mathcal{H}, \delta/N_{\mathrm{nt}})$ this concludes the proof.

For the general situation, assume that there are $N_{\mathrm{nt}}$ non-trivially and $N - N_{\mathrm{nt}}$ trivially generalizing objectives. If $M = 0$, then $\mathcal{L}_i(h) = \widehat{\mathcal{L}}_i(h)$ for all $i = 1, \ldots, N$ and for all $h \in \mathcal{H}$. Then, Pareto-optimal and empirically Pareto-optimal sets coincide, and $\hat{h}^* = h^*$ fulfills the statement of the theorem.

Otherwise, assume without loss of generality that the objectives are ordered such that, $\mathcal{C}_i(n_i, \mathcal{H}, \delta/N_{\mathrm{nt}}) > 0$ for $i \in [N_{\mathrm{nt}}]$ and $\mathcal{C}_i(n_i, \mathcal{H}, \delta/N_{\mathrm{nt}}') = 0$ for $i \in \{N_{\mathrm{nt}} + 1, \ldots, N\}$. Let $\mathcal{G} = \{h \in \mathcal{H} : \widehat{\mathcal{L}}_i(h) = \widehat{\mathcal{L}}_i(\hat{h}^*)$ for $i \in \{N_{\mathrm{nt}} + 1, \ldots, N\}\}$. Note that also $\mathcal{G} = \{h \in \mathcal{H} : \mathcal{L}_i(h) = \mathcal{L}_i(h^*)$ for $i \in \{N_{\mathrm{nt}} + 1, \ldots, N\}\}$, because $\mathcal{L}_{N_{\mathrm{nt}}+1}, \ldots, \mathcal{L}_N$ are trivially generalizing. $\mathcal{G}$ is a subset of $\mathcal{H}$ that is non-empty (because $\hat{h}^* \in \mathcal{G}$). Consequently, the inequalities of Lemma 1 and Theorem 2 hold also as statements for all $g \in \mathcal{G}$ rather than $h \in \mathcal{H}$. Because $\hat{h}^*$ is empirically Pareto-optimal within $\mathcal{H}$ with respect to $\widehat{\mathcal{L}}_1, \ldots, \widehat{\mathcal{L}}_N$, it is also empirically Pareto-optimal in $\mathcal{G}$ with respect to $\widehat{\mathcal{L}}_1, \ldots, \widehat{\mathcal{L}}_M$. Applying the result from the case without trivially-generalizing objectives to this situation, we obtain that there exists $h^* \in \mathcal{G}$ such that for all $i \in [N_{\mathrm{nt}}]$

$$\mathcal{L}_i(\hat{h}^*) \leq \mathcal{L}_i(h^*) + \mathcal{C}_i(n_i, \mathcal{H}, \delta/N_{\mathrm{nt}}) \tag{26}$$

For $i \in \{N_{\mathrm{nt}} + 1, \ldots, N\}$, we have $\mathcal{L}_i(\hat{h}^*) = \mathcal{L}_i(h^*)$, because $h^* \in \mathcal{G}$. Consequently, inequality (26) holds also for these (with $\mathcal{C}_i(n_i, \mathcal{H}, \delta/N_{\mathrm{nt}}) = 0$), which concludes the proof.

