# OpenReview forum: "Generalization In Multi-Objective Machine Learning"
_TMLR — Rejected by TMLR_

### Review · Reviewer_RQtZ · 2022-06-25

**Summary Of Contributions:**

This paper extends the generalization analysis of single-objective machine learning to the setting with multi-objectives. This paper considers two types of bounds. The first is generalization for scalization where the multi-objective is transformed to a single objective. The second is generalization in the pareto excess, where the authors show how the pareto-optimal solution for the empirical objective would relate to the population objective for the risk objective.

**Broader Impact Concerns:**

I do not see any broader impact concerns.

**Requested Changes:**

I would suggest the authors to provide a deeper analysis to get nontrivial generalization bounds in mulbi-objective problems.

I would like to see explanations or examples on ray completeness

Typos:
- Thm 6: $h((x'),y')-(h(x),y)$
- for any $\delta>0$ should be for any $\delta\in(0,1)$

**Strengths And Weaknesses:**

Strength: The paper is well written and easy to follow. The topic is interesting since there is scarce study on the generalization analysis for multi-objectives.

Weakness:
- Most of the results in the paper follow by applying the union bounds on the generalization bounds for single objectives. For example Lemma 1, Theorem 2, Theorem 3 and the first part of Theorem 4 follow directly from the union bounds, which are not surprising.
- The second part of Theorem 4 requires an assumption on the ray-completeness . It seems that this is a strong assumption. The authors do not give explanation and examples for this assumption to hold.

In summary, while this is an initial study on the generalization analysis for multi-objectives, the obtained results are not nontrivial. The analysis is standard.

---

> ### Author Response · Authors · 2022-06-27
> **Not the assessment we had expected from TMLR**
>
> Thank you for your timely review. We agree that the presented results mostly have elementary proofs and we are quite open about this in the manuscript. However, we disagree with the claim that this would make the results trivial. The merit of a result should not be judged by how difficult its proof is, especially in an unexplored research area as is multi-objective generalization. See, for example, our comparison to [Cortes, Mohri, Gonzalvo, Storcheus. "Agnostic learning with multiple objectives", NeurIPS 2020] in Section 5.1: their result states a looser bound but has a more involved proof.
>
> TMLR's evaluation criteria are meant to be *"Are the claims made in the submission supported by accurate, convincing and clear evidence?"* and *"Would at least some individuals in TMLR's audience be interested in knowing the findings of this paper?"* We still believe both is the case for our submission.
>
> Regarding the technical comment, we'll be happy to extend the discussion of *ray completeness* in Section 3. In the situation with two bounded objectives, an condition equivalent to ray completeness is that the Pareto-front is a continuous curve between some point on the $\mathcal{L}_1$-coordinate axis and some point on the $\mathcal{L}_2$-coordinate axis, except the origin. An example of such a situation is a classification task in the realizable setting with *classification error* and *computational cost* as objectives. For a sufficiently rich hypothesis set, the smallest achievable error will be a continuous and monotonically decreasing function of the specified computational budget. Consequently, the Pareto-front will be a continuous curve between a point $(a,0)$, where $a$ is the classification error of the classifier with (suitably defined) minimal budget, and a point $(0,b)$, where $b$ is the smallest computational cost for a classifier achieving classification error $0$. Note that without realizability, the curve would still be continuous starting at $(a,0)$, but the end point of the Pareto-curve would not lie on the $\mathcal{L}_2$-coordinate axis. Consequently, *ray completeness* would not be fulfilled. Overall, we expect ray completeness only to hold in special cases, and -more generally- generalization in the form of Theorem 4b to be the exception rather than the rule for real-world settings.

---

### Review · Reviewer_BnNL · 2022-07-07

**Summary Of Contributions:**

This work considers a setting with multiple potential loss functions and associated generalization error bounds. The authors provide generalization bounds for Lipschitz-scalarizations of the loss functions, along with showing that empirically computed Pareto optimal points are close to some Pareto optimal point on the true loss functions over the underlying population distribution.


**Requested Changes:**

Please see above. The only suggestion above that would secure my recommendation is d).

**Strengths And Weaknesses:**

Strengths:
The paper is well written and I did not find any factual mistakes.

Theorem 4 is interesting and I thought the lower bound was interesting.

Weaknesses

a) I find the application of this work somewhat limited. Though the paper does establish that any empirical Pareto optimal hypothesis is close to some true Pareto optimal hypothesis (under hypothesis), it fails to address the fundamental question of *how* to find this Pareto optimal point, or the frontier of such points. It’s not particularly surprising that the empirical Pareto frontier converges to the true one - however it’s much harder to learn that frontier, or even find one point on that frontier.

b) I find the implication of Theorem 3 somewhat confusing. From inspection of the proof, it does not use the fact that $h^{\ast}$ is Pareto optimal - indeed, the result is true for any $h^{\ast}$. Can the authors clarify?

c) With the exception of Theorem 4 the results and techniques given are somewhat straightforward based on Lemma 1. It would be nice if the paper got to the substance - i.e. Theorem 4 and Theorem 5 faster. Indeed, I could imagine skipping Theorem 2 and indeed providing the construction of Theorem 5 in the body.

d) I like section 4 Applications - the paper would benefit greatly if the ideas in this paper were applied to one of these settings and an improved result were given or an existing result were reproduced.

e) Cortes’20 seems to present an algorithmic improvement as well to handle any linear secularization. Maybe the authors could comment on that as well? Is there room for algorithmic improvement?

My final comment is that I do think the ideas in this paper are interesting but I would like to see a larger contribution and scope of this paper.

---

> ### Author Response · Authors · 2022-07-15
> **Response to Reviewer BnNL**
>
> Thank you for the detailed review. In the following we clarify some aspects.
>
> **the paper [...] fails to address the fundamental question of how to find this Pareto optimal point, or the frontier of such points**
>
> This might be a misunderstanding. The literature contain is a large number of existing multi-objective optimization techniques that can be used to find *empirically* Pareto-optimal hypotheses. We discuss this *optimization* aspect at the end of Section 2 and related learning methods in Section 5, but it is orthogonal to our contribution in this manuscript.
>
> The topic of our work is to establish in which sense and under which assumptions the solutions found by such a optimization methods are also (approximately) Pareto-optimal with respect to the true objectives. This aspect of *generalization* is completely underexplored in the literature. Our work is the first one to formulate even the most fundamental principles. However, only if optimization and generalization come together, then learning from data is actually justified (=the fundamental theorem of statistical learning).
>
> **It’s not particularly surprising that the empirical Pareto frontier converges to the true one**
>
> That would be the common belief, but *it is actually not correct*. One result in our work is that  empirically Pareto-optimal hypotheses can be far from actually Pareto-optimal! We believe that this is a *surprising* result that the community should know about.
> For generalization in the Pareto-curve sense to hold, additional assumptions are required, where the one we identified is quite strong and usually not fulfilled. We’ll be happy to emphasize this aspect more in the manuscript.
>
> **Theorem 3 [...] the proof [...] does not use the fact that $h^\*$ is Pareto optimal  - indeed, the result is true for any $h^\*$**
>
> That is correct. Theorem 3’s statement is a special form of approximate dominance, and because every hypothesis $h$ is dominated by a Pareto-optimal one, the statement also holds for $h$ itself. We can rephrase the Theorem in this way, if that makes it less confusing. Originally we didn’t do so because we wanted to keep the structure consistent with the following theorems, and because it’s less surprising a result for general hypotheses.
>
> **Indeed, I could imagine skipping Theorem 2 and indeed providing the construction of Theorem 5 in the body.**
>
> We will create a revision of the manuscript that includes the proof of Theorem 5 in the main body. We would prefer to keep Theorem 2, as it is of independent value and also used elsewhere, e.g., in the proof of Theorem 6.
>
> **the paper would benefit greatly if the ideas in this paper were applied to one of these settings and an improved result were given or an existing result were reproduced.**
>
> Please see also our reply to Reviewer cYX2. We had meant the comparison to (Cortes et al, 2020) as such a demonstration. We will upload a revised manuscript with more concrete examples.
>
> **Cortes’20 seems to present an algorithmic improvement as well to handle any linear secularization. [...] Is there room for algorithmic improvement?**
>
> (Cortes et al, 2020) uses a specific scalarization, which allows for an efficient optimization algorithm. Our result establishes a tighter bound for the found solution. However, the improved terms are hypothesis-independent, therefore the optimization algorithm would not change.

---

### Review · Reviewer_cYX2 · 2022-07-12

**Summary Of Contributions:**

This work considers the generalization of multi-objective losses.
They provide generalization bounds over scalarizations: Theorem 2.
Then they provide a theorem that says under some assumption, an empirical Pareto optimal can be close to a Pareto optimal solution. Also, they show that if they did not use any assumption the result is not anymore true.

**Broader Impact Concerns:**

no ethical implications.

**Requested Changes:**

1. Definition 2: strictly-> strongly.
2. Ray complete needs additional intuition/discussion.
3.The footnote 3. needs some more explanation and intuition and exploring the weaker assumption would strengthen this work.
4. All the footnotes need to be in the main body. I do not find it useful for them to be footnotes.
5. Weakly dominates from definition 2, is not used anywhere. I do not why the authors defined it if it isn't used anywhere.
6. About the applications: Some parts are vague there, and it seems more like a list of papers. For instance, for the regularization, the second paragraph is the basic definition and the ideas of a regularizer. For me it should be better to have less applications and more concrete ideas of how the results of these work apply to them. Otherwise, some paragraphs in the intro instead of a full section would suffice.
7. Finally, in the conclusion: Characterization is a strong word in my opinion for the second result as it is based on an assumption and the hardness result does not show that this assumption is required.

**Strengths And Weaknesses:**

1. This work is well written and all the claims are accurate.
2. This work provide some interesting properties about multi-objective optimization.
3. On the downside, most of the proofs are quite easy: Theorems 2,3 and 5, basically the proofs follow basic steps from any work that provides single objective generalization bounds. Also, theorem 2 is something well known under the assumption of point-wise convergence of all coordinates.
4. The authors provide several applications of their work to show why they consider their results important.

---

> ### Author Response · Authors · 2022-07-15
> **Response to Reviewer cYX2**
>
> Thank you for your review, in the following we clarify some aspects.
>
> **On the downside, most of the proofs are quite easy**
>
> We agree that the theorems you mention mostly have elementary proofs, but we do not agree that this would constitute a weakness. It is the statements of the theorem that should be judged, not how difficult their proofs are. Please see also our response to Reviewer RQtZ.
>
> **For me it should be better to have less applications and more concrete ideas of how the results of these work apply to them**
>
> Thank you for the helpful feedback. The purpose of the section was not to illustrate specific examples but to highlight the broad range of situations that so far have been studied in isolation rather than in a unified way, and whose multi-objective aspect has been ignored. As concrete example we meant the comparison to (Cortes et al, 2020) of Section 5.1.
> In light of your comments we will upload a revision of the manuscript that provides more examples of such concrete applications.
>
> **The footnote 3. needs some more explanation and intuition and exploring the weaker assumption would strengthen this work**, **Ray complete needs additional intuition/discussion.**
>
> We will expand the discussion and give concrete examples, see also our response to Reviewer RQtZ.  Note, however, that the conditions are quite restrictive, even the weaker one. In general we expect generalization of the Pareto-curves *not to hold*.
>
> **other requested changed**
> Thank you for the remarks, we will revise the manuscript accordingly.

---

### Decision · Action_Editors · 2022-08-07

**Recommendation:** Reject

**Comment:**

This manuscript has the potential to be of interest to the TMLR community. However, in its current form, it does not sufficiently focus on the non-trivial parts of the contribution. The authors are encouraged to consider the reviewers' comments and suggestions, and to submit a new version that focuses on the more interesting aspects of the work, while removing or deferring to an appendix the more straightforward observations.